# Guess-and-Learn: Benchmarking the Cumulative Error of Cold-Start Adaptation

**Roland Arnold** *roland.w.arnold@gmail.com*
*Independent Researcher*

**Reviewed on OpenReview:** *https://openreview.net/forum?id=uNxKhjcRp9*

## Abstract

Evaluation of machine learning models typically emphasizes final accuracy, overlooking the cost of adaptation: the cumulative errors incurred while learning from scratch. Guess-and-Learn (G&L) v1.0 addresses this gap by measuring cold-start adaptability—the total mistakes a model makes while sequentially labeling an unlabeled dataset. At each step, the learner selects an instance, predicts its label, receives the ground truth, and updates parameters under either online (per-sample) or batch (delayed) mode. The resulting error trajectory exposes adaptation speed, selection quality, and bias—dynamics invisible to endpoint metrics. G&L defines four tracks (Scratch/Pretrained × Online/Batch) to disentangle the effects of initialization and update frequency. We formalize the protocol, relate it to classical mistake-bound theory, and estimate a heuristic "oracle reference band" for MNIST as a plausibility reference. Baseline experiments on MNIST and AG News, spanning classical methods (Perceptron, $k$-NN), convolutional architectures (CNN, ResNet-50), and pretrained transformers (ViT-B/16, BERT-base), reveal systematic differences in early-phase efficiency: smaller models can adapt with fewer initial errors, while pretraining benefits vary by domain. Across settings, current models remain well above the oracle band, highlighting an adaptability gap. By quantifying the mistake cost of early learning, G&L complements conventional benchmarks and provides a reproducible framework for developing learners that are not only accurate in the limit but also reliable from the first examples.

## 1 Introduction

Modern machine learning systems achieve high accuracy, but this success often depends more on scale than adaptability. Most models require large amounts of labeled data, learn slowly from scratch, and adapt poorly to new domains where in-domain labels are scarce. In contrast, truly efficient learners would achieve competence with fewer examples and recover more quickly under distribution shift.

Evaluation practice, however, is dominated by two metrics: final accuracy on a held-out test set and label efficiency (measuring how many labeled samples are needed to reach a target performance). While useful, both overlook the cost of adaptation—the total mistakes a model makes while learning. Two models may reach similar accuracy but differ drastically in how many errors they incur along the way.

Guess-and-Learn (G&L) addresses this gap by treating cumulative error, not label count, as the primary metric. At each step, a learner selects an instance, predicts its label, receives the ground truth, and updates parameters. The benchmark tracks cumulative mistakes over the full labeling process, producing an error trajectory that directly reflects adaptation speed and efficiency. This perspective is especially relevant in domains where early mistakes carry real costs: degraded user experience in interactive systems Amershi et al. (2014), increased safety risk in autonomous systems, or wasted resources when labels are expensive to obtain.

Our central question is: *How many total errors does a model make when labeling every instance in a dataset, starting from zero in-domain knowledge?* Formally, at step $t$, the cumulative error is

$$E_t = \sum_{i=1}^{t} \mathbf{1}\{\hat{y}_i \neq y_i\}, \tag{1}$$

where $\hat{y}_i$ is the model's prediction, $y_i$ the ground truth, and $\mathbf{1}\{\cdot\}$ the indicator function.

For humans—or an idealized clustering-and-reasoning oracle—the answer can be remarkably low. On MNIST, probabilistic analysis Mitzenmacher & Upfal (2017) suggests a plausible floor in the single digits (see Appendix C), and under realistic assumptions accounting for ambiguous instances Yadav & Bottou (2019), mastery is estimated to be possible after fewer than twenty errors. In practice, even state-of-the-art models make tens or hundreds of times more mistakes, underscoring a clear adaptability gap.

G&L is not a replacement for existing benchmarks, but a complementary diagnostic. By quantifying the mistake cost of early learning, it provides a concrete, reproducible metric for assessing model adaptability and for driving the development of learners that are not only accurate in the limit but also reliable from the very first examples.

**Contributions.** This paper's contributions are:

- **A new evaluation protocol (G&L)** for measuring cold-start adaptation, defined by four tracks that isolate initialization and update frequency.
- **A heuristic oracle reference band** for MNIST to provide a plausible performance floor for early-phase errors.
- **A reproducible experimental framework,** including open-source code[1] with a pinned commit and fixed seeds.
- **An analysis of training policy effects** (e.g., update cadence, weight resets) that highlights a capacity-agility trade-off in early adaptation.

## 2 Purpose & Scope

The goal of Guess-and-Learn (G&L) is not simply to produce another leaderboard, but to establish a benchmark that makes early-phase adaptability measurable. By quantifying the mistakes made during cold-start learning, G&L turns variation in efficiency into a concrete, comparable target for model improvement.

**Cold-start setting.** In G&L, the learner begins with:

- Zero in-domain labeled instances.
- Optional task-agnostic prior knowledge (e.g., pretrained weights).
- A requirement to predict a label before receiving the ground truth.
- No option to abstain from prediction.
- Freedom to select the next instance to label by any strategy.

These conditions hold throughout. Any form of manual seeding, few-shot prompting, or oracle hints constitutes a warm start and is outside the scope of G&L v1.0.

**Protocol.** The protocol involves a sequential, full-pool labeling process. At each step, the learner (i) selects an unlabeled example from the pool, (ii) predicts its class, (iii) receives the ground-truth label from an oracle, and (iv) updates its parameters based on a fixed schedule. Every prediction contributes to cumulative error. The primary performance measure is the final cumulative error, $E_N$, after labeling a pool of $N$ instances.

---

[1]Available at `https://github.com/RolandWArnold/guess-and-learn-benchmark` (Commit: `5653338`).

**Tracks.** G&L v1.0 defines four tracks that isolate the influence of initialization and update frequency:

- **Scratch–Online (SO):** random initialization; update after every labeled example.
- **Scratch–Batch (SB):** random initialization; update after batches of size $K > 1$.
- **Pretrained–Online (PO):** initialized from pretrained weights; update after every labeled example.
- **Pretrained–Batch (PB):** initialized from pretrained weights; update after batches of size $K > 1$.

These tracks separate the influence of prior knowledge (scratch vs. pretrained) from the influence of update schedule (per-step vs. batch).

**Oracle reference band (MNIST).** To contextualize MNIST results, we include a heuristic oracle band: an estimate of the minimum plausible number of errors under idealized assumptions (e.g., perfect clustering with immediate generalization after one labeled example per class). Its width reflects uncertainty from factors such as class imbalance, intra-class variation, and label noise Northcutt et al. (2021). This heuristic estimate is not a formal bound but an illustrative reference, included to highlight the large gap between current systems and efficient learning under ideal conditions (Appendix C).

## 3 Relation to Existing Paradigms

Guess-and-Learn (G&L) borrows familiar settings from existing paradigms but alters the objective to measure the cumulative error cost of cold-start adaptation.

**Active learning.** Pool-based active learning seeks to minimize label queries needed to reach a target accuracy Cohn et al. (1994); Settles (2009). G&L uses the same interactive pool-selection setting but inverts the objective: it measures the total mistakes made while labeling the entire pool. Two reference points are useful: (i) a random baseline: for $C$ balanced classes, uniform guessing incurs $\mathbb{E}[E_N] = N(1 - 1/C)$; and (ii) an existence floor: at least $C - 1$ mistakes are unavoidable in the worst case, since the learner must encounter at least one example of each class before knowing it exists Angluin (1988). The combination of pool-based selection with sequential model updates is often termed Interactive Machine Learning Amershi et al. (2014); Fails & Olsen Jr (2003), where the learner's goal is to minimize the cost of the interaction (cumulative errors) rather than just the final generalization error.

**Online learning.** The G&L Scratch–Online (SO) track is a direct empirical analogue of the mistake-bound model Littlestone (1988). For linearly separable data with margin $\gamma$ and diameter $R$, the Perceptron has the classic bound $E_N \leq R^2/\gamma^2$ Novikoff (1962); Littlestone (1988). G&L-SO instantiates this setting on modern, high-dimensional data, with the added twist that the learner actively selects the sequence of examples.

**KWIK learning (abstention).** "Knows What It Knows" frameworks allow abstention until confidence is high, thereby avoiding mistakes Li et al. (2008). G&L forbids abstention; every instance must be predicted. This models mandatory-decision scenarios (e.g., medical triage, autonomous navigation) and ensures the full cost of uncertainty is counted.

**Curriculum & self-paced learning.** Acquisition strategies impose a curriculum Bengio et al. (2009); Kumar et al. (2010). Uncertainty sampling tends toward "hard-first", while confidence sampling is "easy-first". Heuristically, if a strategy maintained per-step margin $\gamma_t \geq \gamma_{\min}$, the Perceptron bound tightens to

$$E_N \leq \frac{R^2}{\gamma_{\min}^2}$$

This motivates margin-aware sampling, though it is not a guarantee for the heuristics we evaluate.

**Prequential evaluation.** G&L's predict-then-update loop mirrors prequential evaluation in data streams Dawid (1984); Gama et al. (2013), with three differences: (i) a fixed pool in cold-start; (ii) cumulative error is the primary metric; and (iii) the learner controls instance order.

## 4 Methodology

### 4.1 G&L Protocol

At each step the learner:

(i) selects one unlabeled instance using a chosen acquisition strategy;

(ii) predicts its label;

(iii) receives the ground-truth label from an oracle;

(iv) updates its parameters according to a predefined schedule.

Predictions are mandatory (no abstention), and errors are accumulated over time. The primary metric is final cumulative error $E_N$ after all $N$ instances are labeled.

### 4.2 Benchmark Tracks

We cross two factors—initialization and update schedule—to form four tracks (Table 1). Online tracks update after each instance; batch tracks update after groups of size $K > 1$ (default $K = 50$ unless otherwise specified).

Table 1: G&L tracks formed by the cross of initialization and update schedule.

| Track | Initialization | Update Schedule | Batch Size ($K$) |
|-------|----------------|-----------------|------------------|
| G&L–SO | Scratch | Online | 1 |
| G&L–SB | Scratch | Batch | $> 1$ (default $K$=50) |
| G&L–PO | Pretrained | Online | 1 |
| G&L–PB | Pretrained | Batch | $> 1$ (default $K$=50) |

### 4.3 Acquisition Strategies

We evaluate six established strategies:

- **Random:** uniform sampling without replacement.
- **Confidence (easy-first):** select $x$ maximizing $\max_y P_\theta(y \mid x)$ Bengio et al. (2009).
- **Least-confidence:** select $x$ that minimizes $\max_y P_\theta(y \mid x)$ Lewis & Gale (1994).
- **Margin:** select $x$ with smallest gap $p_1 - p_2$ between the top two class probabilities Schein & Ungar (2007); Tong & Koller (2001).
- **Entropy:** select $x$ with largest $-\sum_y P_\theta(y \mid x) \log P_\theta(y \mid x)$ Settles (2009).
- **K-Center Greedy:** selects $x$ such that the maximum distance from any unlabeled point to its nearest labeled example is minimized Sener & Savarese (2018).

These are interpretable baselines to reveal how acquisition logic shapes the error trajectory; we do not claim optimality for G&L.

### 4.4 Implementation Details

All models produce class-probability vectors $P_\theta(y \mid x)$ with prediction $\hat{y} = \arg\max_y P_\theta(y \mid x)$. Models producing logits use a softmax Bridle (1990); $k$-NN probabilities derive from distance-weighted neighbor votes (normalized) Cover & Hart (1967). For $k$-NN, 'learning' consists of adding each newly labeled instance to the pool of reference examples used for subsequent distance-weighted predictions. Ties (in acquisition or prediction) are broken uniformly at random. The pool is processed without replacement until exhaustion. Unless noted, results are averaged over fixed seeds.

## 4.5 Experimental Setup

**Datasets.** MNIST (image classification) LeCun et al. (1998) and AG News (text classification) Zhang et al. (2015). For G&L, the unlabeled pool is drawn from each dataset's official test set and processed without replacement, solely to standardize the unlabeled pool for measuring adaptation cost; we do not report conventional test accuracy here, so this does not create leakage for our target metric.

**Models.** *Vision (MNIST): $k$-NN ($k$=7)* Cover & Hart (1967); Fix & Hodges (1951), Perceptron Rosenblatt (1958), a small 3-layer CNN LeCun et al. (1998), ResNet-50 He et al. (2016), ViT-B/16 Dosovitskiy et al. (2021); Vaswani et al. (2017). *Text (AG News): $k$-NN*, linear Perceptron, BERT-base Devlin et al. (2019); Vaswani et al. (2017).

**Pretrained tracks.** In PO (Pretrained–Online), pretrained backbones are used as frozen feature extractors with a newly initialized linear head trained online. In PB (Pretrained–Batch), all parameters are fine-tuned end-to-end on the batch schedule.

**Acquisition.** All six strategies from 4.3 are used.

**Updates.** Online tracks update every step; batch tracks use $K > 1$ (default $K$=50). For our ablation analysis, we vary this parameter across $K \in \{10, 50, 200\}$.

**Reporting.** Unless noted, results are averaged over seeds and reported for $n$=300 early-phase subsets; full-pool curves are included for MNIST to contextualize the oracle band.

**Hyperparameter tuning.** To reflect realistic cold-start usage, hyperparameters (e.g., learning rates, architecture depth) were selected based on standard conventions for final accuracy convergence rather than explicitly tuned to minimize cumulative error. This focuses the evaluation on the adaptability of standard architectures as they are typically instantiated.

# 5   Results and Analysis

We organize findings into four themes: model adaptability, strategy effects, training policies, and cost–performance trade-offs.

## 5.1   The Adaptability Gap

Most models are far from optimal in early adaptation. On MNIST, the best-performing ViT-B/16 track—pretrained-batch (PB)—accumulated $136.0 \pm 4.75$ errors at $n = 300$ under Random acquisition (mean $\pm$ standard error across seeds), a finding visualized in Figure 1. In contrast, a conceptual clustering oracle—requiring only one representative example per class—suggests a 7–12 mistake plausibility band under idealized assumptions (see Appendix C). We note that datasets with significant label noise would inherently possess higher error floors, though the magnitude of the current adaptability gap suggests that substantial room for improvement remains regardless of this floor. For AG News, whose class boundaries are more ambiguous (e.g., *World* vs. *Business*), and where prior work reports nontrivial disagreement and label noise Northcutt et al. (2021), we do not provide an oracle estimate.

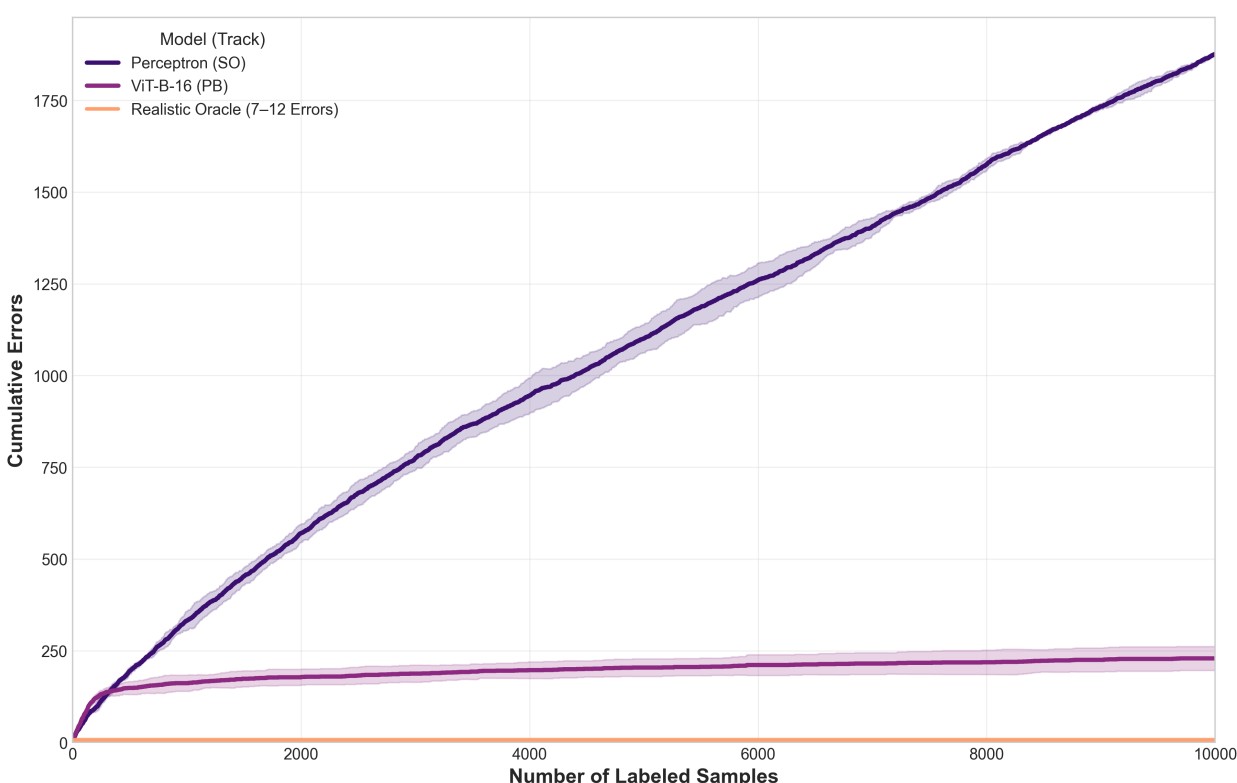

Figure 1: MNIST cumulative error trajectories under Random acquisition. We compare the Perceptron (SO) and ViT-B/16 (PB) against a heuristic oracle band (7–12 errors). All models remain well above the band during early adaptation, highlighting the adaptability gap.

## 5.2   Capacity vs. Agility

Adaptability is not a simple function of size. In cold-start, smaller online learners can outperform larger models early on (Figure 2). On the first 300 MNIST samples, the Perceptron incurs fewer cumulative errors than both a CNN and $k$-NN. High capacity (ResNet-50, ViT-B/16) does not immediately translate into efficient boundary formation when labeled data are scarce.

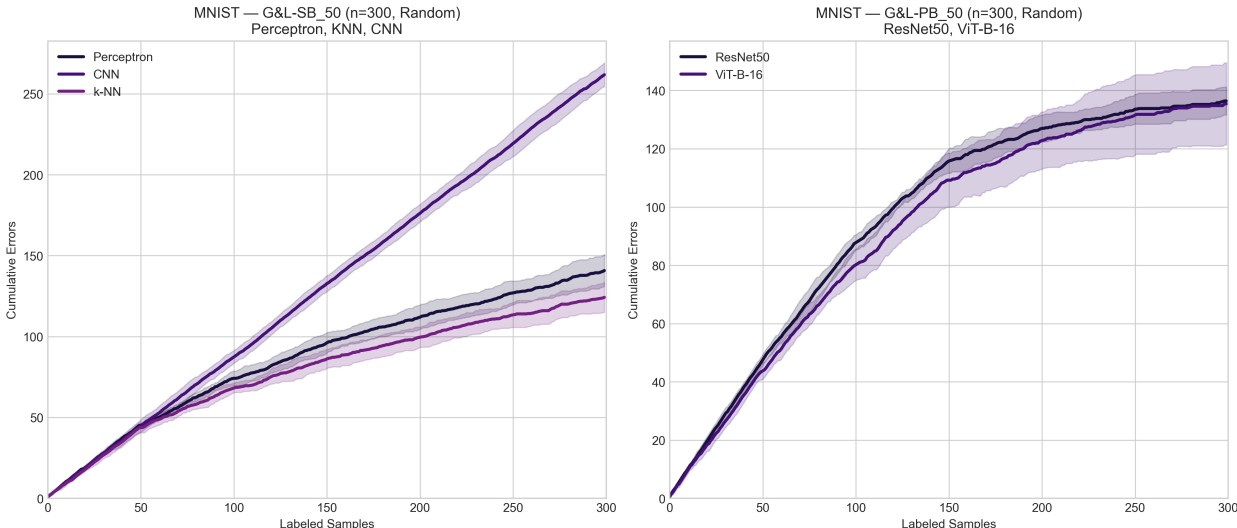

Figure 2: Capacity vs. agility on MNIST ($n$=300, Random acquisition). Left: small SO models (Perceptron, $k$-NN, CNN). Right: pretrained PB$_{50}$ models (ResNet-50, ViT-B/16). Smaller online learners adapt faster.

## 5.3 Impact of Acquisition Strategies

Effects are task- and model-dependent, as shown in Figure 3. On MNIST with a CNN (SO), *Confidence* (easy-first) yields the lowest cumulative error over the first 300 samples. Uncertainty-based strategies (Entropy, Margin, Least-Confidence) underperform even Random sampling, likely because they front-load ambiguous cases that inflate early mistakes. On AG News with BERT-base (PO), strategy differences are small and often within variance bands in the first 300 samples; strong priors dominate.

However, geometric sampling proves highly effective in specific regimes: as shown in Table 2, K-Center Greedy reduces cumulative error by 15.9% for $k$-NN on MNIST and 11.7% for ResNet-50 compared to random sampling. This highlights a clear "geometric win" for models that rely heavily on feature-space distance, though the advantage diminishes for text models (AG News) where density-based uncertainty often prevails.

Table 2: Maximum potential error reduction: Comparing the best-performing strategy (post-hoc) against the random baseline for each track. This highlights the upper bound of efficiency gains available with current methods. **Min Error** reports the cumulative error count ($n = 300$). Green indicates error reduction (improvement); red indicates degradation.

| Dataset | Model | Track | Best Strategy | Min Error | Random | $\Delta$ Error (%) |
|---------|-------|-------|---------------|-----------|--------|--------------------|
| AG News | BERT-base | G&L-PB$_{50}$ | Confidence | $84.8 \pm 11.8$ | $101.2 \pm 8.9$ | −16.2 |
| AG News | Text $k$-NN | G&L-SB$_{50}$ | Confidence | $128.4 \pm 11.5$ | $138.6 \pm 9.2$ | −7.4 |
| AG News | Text Perceptron | G&L-SO | Confidence | $141.8 \pm 9.9$ | $189.0 \pm 4.3$ | −25.0 |
| MNIST | $k$-NN | G&L-SB$_{50}$ | K-Center Greedy | $104.4 \pm 7.3$ | $124.2 \pm 9.3$ | −15.9 |
| MNIST | CNN | G&L-SO | Confidence | $69.2 \pm 6.6$ | $92.8 \pm 8.4$ | −25.4 |
| MNIST | Perceptron | G&L-SO | Confidence | $98.8 \pm 4.4$ | $143.2 \pm 6.4$ | −31.0 |
| MNIST | ResNet-50 | G&L-PO | K-Center Greedy | $238.2 \pm 7.7$ | $269.8 \pm 4.3$ | −11.7 |
| MNIST | ViT-B/16 | G&L-PB$_{50}$ | K-Center Greedy | $150.4 \pm 3.4$ | $135.4 \pm 14.1$ | +11.1 |

## 5.4 Protocol Validation and Ablations

We performed ablation studies to validate the robustness of the G&L protocol and quantify the impact of update frequency and model persistence.

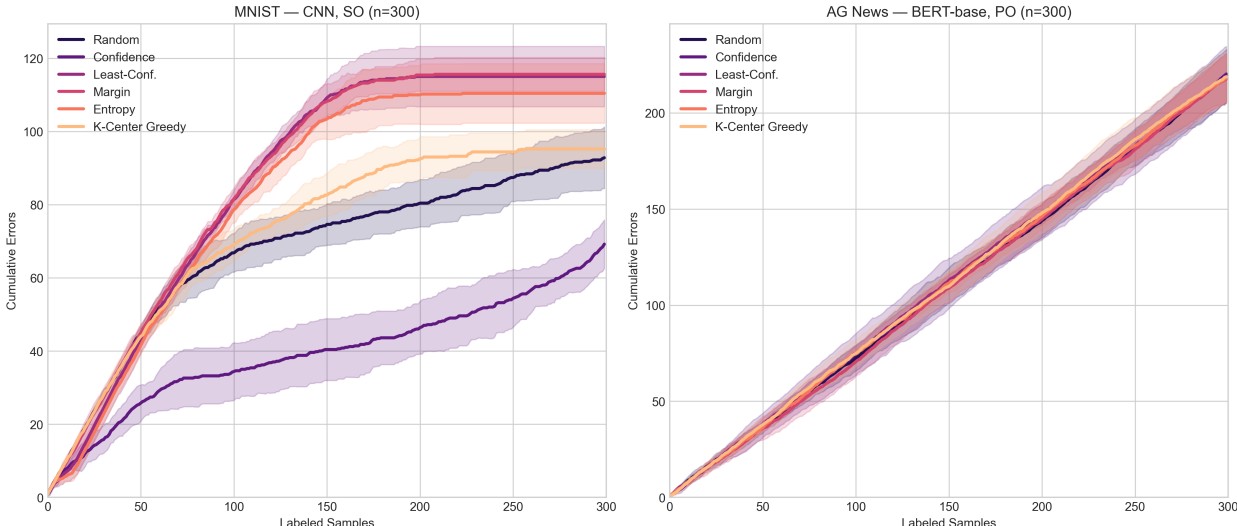

Figure 3: Strategy effects. Left: MNIST—CNN, SO ($n$=300): Confidence (easy-first) yields the lowest cumulative error; entropy/margin/least-confidence are worse than Random. Right: AG News—BERT-base, PO ($n$=300): curves are tightly clustered; strategy choice has minimal impact over the first 300 samples.

**Update Cadence ($K$).** We analyzed the trade-off between computational cost and adaptation speed by varying the batch size $K \in \{10, 50, 200\}$ (Figure 4, Left). While more frequent updates ($K = 10$) generally reduce cumulative error by allowing the model to react faster to mistakes, the relationship is non-monotonic; $K = 200$ unexpectedly outperforms the intermediate $K = 50$. This suggests a trade-off where improved gradient estimation from larger batches can partially offset the penalty of delayed feedback.

**Impact of Continuous State (Weight Reset).** To verify that G&L measures true adaptation rather than repeated independent fitting, we evaluated a "forgetful" baseline that resets the model to its original pretrained weights before every batch (Figure 4, Right). As expected, discarding the learned state significantly degrades performance. This negative control serves as a sanity check to confirm that the benchmark isolates the contribution of the update steps.

## 5.5 Cost–Performance Trade-offs

Plotting final error against mean wall-clock time ($n$=300) reveals distinct cost–performance frontiers by dataset and track. Each point plots the mean performance of a specific model–and–strategy combination; plots are faceted to avoid mixing regimes.

On MNIST, SO models define the most efficient frontier. $k$-NN achieves the lowest error ($\sim$80–100) at moderate cost, while the Perceptron is faster but higher-error; the CNN performs poorly in SO. Deferring updates in SB degrades performance. Pretrained vision models (ResNet-50, ViT-B/16) in PO/PB are costly and not competitive on this dataset.

On AG News, pretrained models are essential. In scratch tracks, Text $k$-NN is a reasonable baseline and outperforms the fast but error-prone Text Perceptron. BERT-base in PB (fine-tuning) attains the lowest errors ($\sim$80–130) at the highest cost; PO (frozen features) is weaker. Aggregating across strategies shifts frontiers (e.g., SO on MNIST toward $k$-NN).

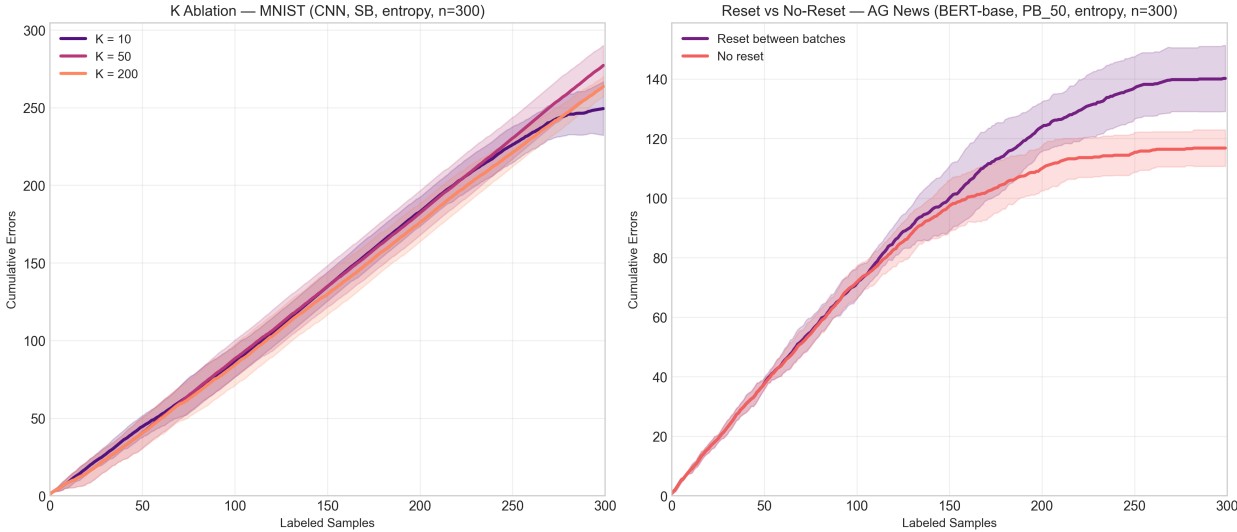

Figure 4: Ablations. Left: An ablation on the batch size $K$ for a CNN on MNIST (SB track, Entropy acquisition). The impact of update frequency is non-monotonic; while more frequent updates ($K = 10$) are most effective, a very large batch size ($K = 200$) unexpectedly outperforms an intermediate one ($K = 50$). Right: AG News BERT-base, $PB_{50}$ Entropy (resetting between batches increases errors).

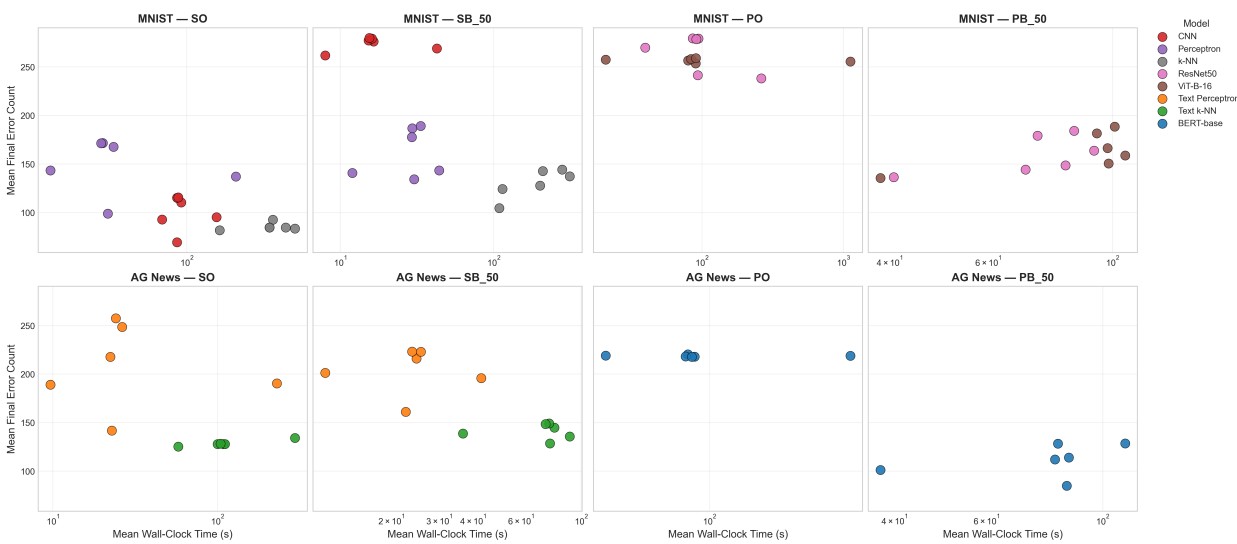

Figure 5: Cost–performance ($n$=300): mean wall-clock time (log-scale) vs. mean final error, faceted by dataset (MNIST, AG News) and track (SO, SB, PO, PB). Each point represents the mean performance of a specific model–and–strategy pairing.

# 6 Discussion

## 6.1 Key Findings and Interpretations

**Adaptability remains challenging.** Even powerful pretrained models accumulate hundreds of early mistakes on simple tasks and remain far above the plausible 7–12 error band on MNIST.

**Agility can outweigh capacity.** On simpler tasks and in early stages, smaller online models (e.g., Perceptron) often incur fewer errors than high–capacity architectures, underscoring a capacity–agility tension. The substantial cumulative error observed in high–capacity models during early adaptation on clean, highly structured benchmarks such as MNIST suggests that this inefficiency reflects structural properties of current large–scale architectures and their training regimes rather than merely dataset noise. If heavily parameterized models struggle to leverage sparse, sequential feedback under controlled, low–noise conditions, it is unlikely that this limitation disappears solely as tasks become more complex. Importantly, there is no theoretical justification to assume that rapid, error–efficient adaptation must inherently trade off against ultimate representational capacity.

**Strategy is context-dependent.** Uncertainty-based methods can hurt when they front-load ambiguous cases (e.g., MNIST with CNN), yet matter little when strong priors dominate (e.g., BERT on AG News).

**Training policies are critical.** Update frequency and weight retention substantially affect cumulative error; retaining learned information between batches is especially important.

**Cost–performance is regime-specific.** No single model excels across all tracks. The value of pretraining and sensitivity to batch updates are task dependent, arguing for regime-aware evaluation.

## 6.2 From Diagnosis to Design

The diagnostics point to concrete opportunities:

- **Architectures & optimizers** designed for rapid, stable per-sample or small-batch updates.
- **Transfer & meta-learning** aimed at improving cold-start error efficiency, not only final accuracy.
- **Hybrid update schemes** that leverage large-model capacity without sacrificing online agility.
- **Acquisition–update co-design** to optimize instance selection jointly with update dynamics.

## 6.3 Limitations and Future Work

G&L v1.0 prioritizes a clear, reproducible protocol, establishing baselines on curated benchmarks. This focus entails specific scope limits that suggest natural directions for future research:

- **Data scale and noise:** We utilized clean, small, and standard datasets to isolate model adaptability from data quality issues. Real-world data is often larger and noisier, and so while establishing baselines in controlled regimes is a prerequisite for measuring performance in complex environments, reliably scaling G&L to these regimes is a natural direction for future work.
- **Metric incentives:** In regimes with extreme class imbalance, cumulative error tracking can yield degenerate solutions. A "lazy" model that predicts only the majority class may incur lower cumulative error than an agile model that actively explores to find rare instances. Future extensions for such pathological cases may need cost-sensitive metrics (e.g., cost-weighted error) to align with downstream utility.
- **Oracle assumptions:** The protocol currently forces the learner to accept oracle labels as absolute truth. A more advanced "skeptical learner" extension could allow models to flag potential label noise for human review, rewarding the identification of dataset errors rather than penalizing the model for disagreeing with them.
- **Ambiguity awareness:** The current protocol enforces strict single-label classification. However, real-world inputs often contain intrinsic ambiguity or validly belong to multiple categories simultaneously.

Future iterations could support multi-label or probabilistic acquisition strategies, rewarding models that capture this nuance rather than forcing a single hard decision.

- **Generalization:** The protocol measures adaptation efficiency within a fixed pool; it does not explicitly assess post-adaptation performance on out-of-distribution domains or open-set recognition tasks where the label space expands over time.

## 7 Conclusion

Guess-and-Learn addresses a clear gap by measuring the cumulative error cost of learning from a cold-start. By tracking mistakes step-by-step, it captures how quickly a model becomes useful—information that final accuracy and label-efficiency do not provide.

The four-track design (scratch vs. pretrained, online vs. batch) and full error trajectories expose actionable differences in adaptation: sensitivity to initialization and update policy, the variable impact of acquisition strategies, and regime-specific cost–performance trade-offs. These distinctions matter in practice: reducing early errors lowers annotation cost, mitigates risk in safety-critical settings, and improves user experience.

At the same time, results show a persistent adaptability gap. On MNIST, modern models make many more early mistakes than a plausible oracle. Closing this gap requires optimizing for agility as well as for final accuracy. We offer G&L not only as a benchmark but as a design driver: a concrete target and a reproducible way to assess progress toward learners that adapt faster, make fewer mistakes, and become reliable sooner.

## 8 Acknowledgements

We thank the Action Editor and the anonymous reviewers for their constructive feedback, which helped improve this manuscript.

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

## A Experimental Reproducibility

**Environment.** All experiments were run on a single NVIDIA A100 (40 GB) GPU, using PyTorch 2.0.1, Hugging Face Transformers 4.54.0, and Python 3.11.

**Scale.** The analysis is based on 579 independent training runs, which produced 484,300 data records.

**Seeds.** Random seeds $[0, 1, 2, 3, 4]$ were fixed across runs.

**Code and data.** The complete source code, configuration files, and logged results to reproduce this benchmark are publicly available at `https://github.com/RolandWArnold/guess-and-learn-benchmark` (Commit: `5653338`). The repository `README.md` provides step-by-step reproduction instructions.

## B Supplemental Visualizations

**Figure 6** Early-stage adaptability on AG News ($n$=300). Pretrained BERT shows a clear advantage over scratch models, reflecting the benefit of task-aligned priors.

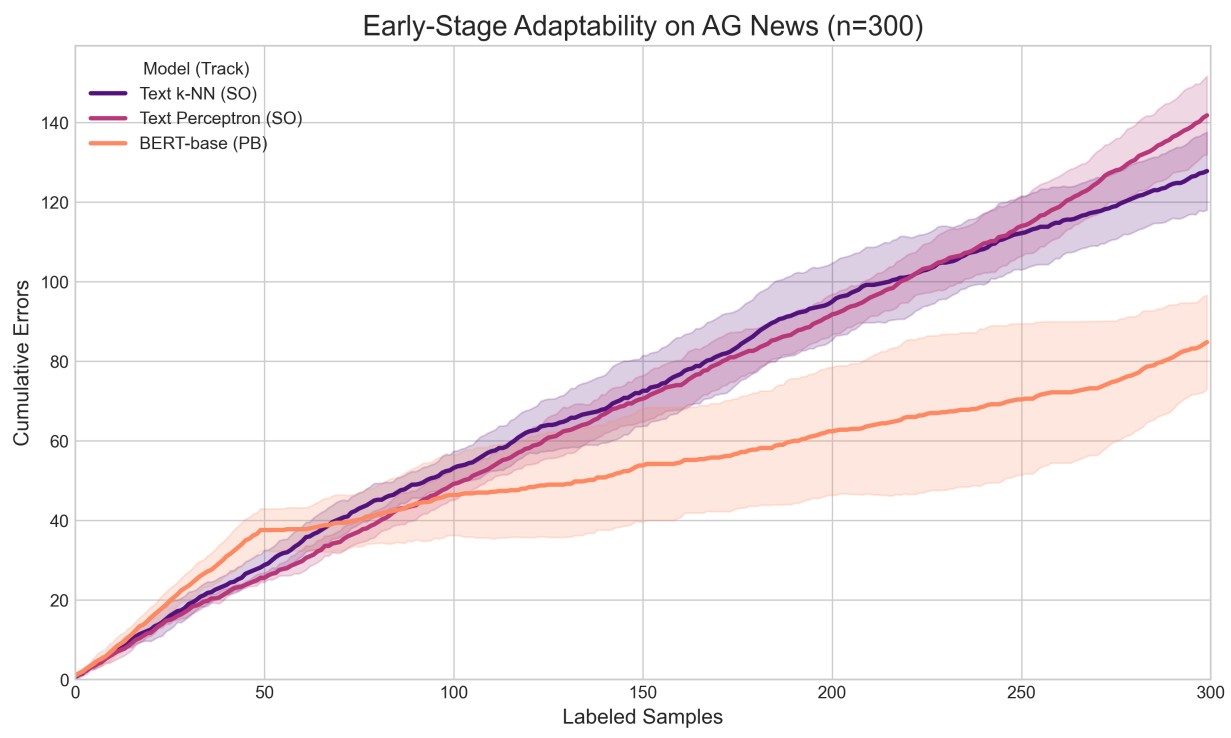

Figure 6: AG News early-stage adaptability ($n$=300).

**Figure 7** Early-stage adaptability on MNIST ($n$=300). The Perceptron adapts quickly and outperforms deeper pretrained models early on, illustrating the capacity–agility trade-off.

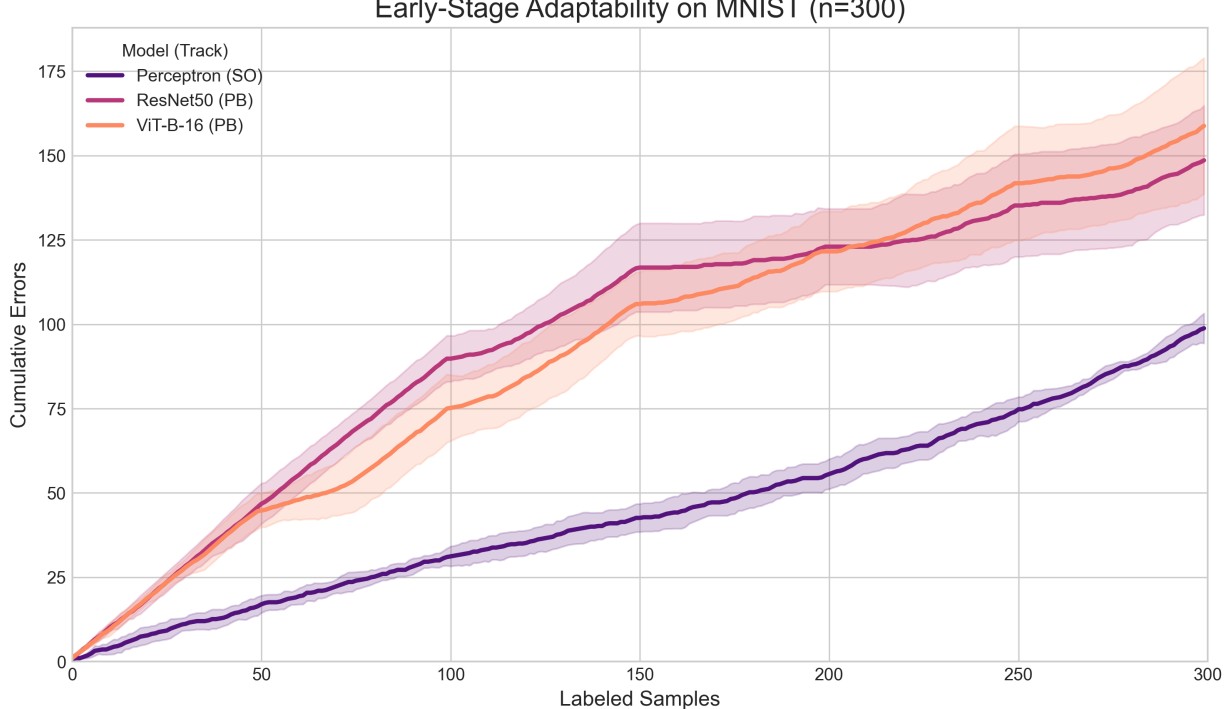

Figure 7: MNIST early-stage adaptability ($n$=300).

## C  Heuristic Oracle Floor for MNIST

We provide a practical, heuristic estimate of a plausible floor on errors for MNIST—the "oracle floor"—based on a simple *cluster-first* oracle and an audit of residual ambiguity. This is an illustrative reference, not a formal bound.

**Cluster-First Oracle**

**Setup.** (i) Embed the unlabeled pool; (ii) partition into $C$=10 clusters (e.g., $k$-means MacQueen (1967); Lloyd (1982); Forgy (1965)); (iii) query one medoid per cluster and assign its label to the cluster.

**Mapping cost.** Associating $C$ abstract clusters to $C$ class names is a coupon-collector variant with feedback Mitzenmacher & Upfal (2017); Motwani & Raghavan (1995). The expected number of mapping errors is

$$\mathbb{E}[E_{\mathrm{map}}] \;=\; C - H_C, \qquad H_C \;=\; \sum_{j=1}^{C} \frac{1}{j}. \tag{2}$$

For MNIST ($C$=10), this gives $\mathbb{E}[E_{\mathrm{map}}] \approx 10 - H_{10} \approx 7.07$.

**Residual ambiguity.** Even with near-perfect clustering, two sources remain:

- *Boundary cost* ($E_{\mathrm{bound}}$): cluster-purity audits suggest $\sim$2–4 borderline cases (e.g., 1 vs. 7, 0 vs. 6).
- *Noise/ambiguity cost* ($E_{\mathrm{noise}}$): label audits identify $\sim$4–6 ambiguous or mislabeled instances (e.g., via Cleanlab; see Northcutt et al. (2021); Yadav & Bottou (2019)).

These overlap with one another and with mapping steps; they are not simply additive.

**Oracle band.** Combining the mapping expectation ($\sim$7 errors) with a small overlapping residual yields a MNIST oracle band of roughly 7–12 errors. This represents a minimum plausible range under idealized conditions, with remaining mistakes attributable to intrinsic dataset ambiguity rather than learner deficiencies.

