# OpenReview forum: "Guess-and-Learn (G&L): Measuring the Cumulative Error Cost of Cold-Start Adaptation"
_TMLR — Accepted by TMLR_

### Review · Reviewer_K2nJ · 2025-11-09

**Summary Of Contributions:**

This paper introduces Guess and Learn, a novel and well-motivated evaluation framework for measuring the ‘cold-start’ adaptation cost of machine learning models. The primary contribution is a shift in the evaluation paradigm from traditional final accuracy to cumulative error, defined as the total mistakes a model makes while sequentially learning an entire unlabelled dataset. This metric captures the practical costs of early learning in real world applications. The G&L protocol is rigorously defined through a sequential ‘predict then update’ loop and features an insightful four track design that systematically disentangles the effects of model initialization and update frequency. Through empirical studies on MNIST and AG News, the paper provides key findings, including the existence of a significant "adaptability gap" between current models and a plausible performance floor, and a clear ‘capacity and agility’ trade-off where smaller models can be more efficient in early stages. Overall, the work delivers a reproducible framework that provides a new lens for assessment, urging the community to optimize not just for final accuracy but for rapid and reliable adaptation from the very first examples.\
Key strength:\
1.The paper addresses a critical but overlooked aspect of model evaluation. The concept of "cumulative error cost" is highly relevant for practical applications where early mistakes are costly.\
2.The G&L framework is simple, intuitive, and clearly defined. The four-track system is a major strength, allowing for a nuanced and insightful analysis of how pretraining and update policies separately contribute to adaptation performance.\
3.The experiments are thorough and lead to valuable insights. The concepts of the "adaptability gap" and the "capacity-agility trade-off" are well-supported by the results and provide a new vocabulary for discussing model behavior.\
4.The authors' commitment to providing an open-source framework is commendable. G&L has the potential to become a valuable diagnostic tool and a standard benchmark for the community, driving research in new directions.\
Key weaknesss:\
1.The method was only tested on two clean, standard datasets. We don't know if the results would be the same for larger and more chaotic real-world data.\
2.The process of checking every single data point one-by-one can be very slow and expensive. This makes it difficult to use with the huge datasets and models common today.\
3.The "oracle reference band" used for comparison is an educated guess, not a proven, perfect score. Therefore, the "adaptability gap" is a helpful illustration, but not a precise measurement.\
4.The models' hyperparameters were likely optimized for traditional metrics like final accuracy. These settings may be suboptimal for minimizing cumulative error, meaning the reported performance might not reflect the true potential of these models under the G&L framework.

**Audience:**

Yes

**Audience Explanation:**

This paper offers a fresh and much-needed perspective on model evaluation by focusing on a very practical question: what is the cost of learning from scratch? Instead of just measuring final accuracy, the proposed G&L framework quantifies the cumulative error, the total number of mistakes a model makes as it learns. This is a far more relevant metric for many real-world scenarios where early performance and reliability are critical. Because of this practical focus, the findings will be relevant to a broad TMLR audience, especially those in active learning, transfer learning, and anyone working on benchmarks for more efficient and reliable ML systems.

**Broader Impact Concerns:**

I have no broader impact concerns regarding this work. The paper introduces a new methodology for evaluating machine learning models, not a new application or model with direct societal implications.I think this work does not take any new ethical risks, and a broader impact statement is not necessary.

**Claims And Evidence:**

Yes

**Claims Explanation:**

The paper's strength begins with its clear and rigorous definition of the Guess-and-Learn (G&L) framework. The authors meticulously detail the sequential learning protocol and its four distinct tracks (SO, SB, PO, PB), creating a well-defined and systematic methodology. This clear framework makes the paper's central claims testable and provides a solid foundation for the experimental work that follows.\
Building on this strong foundation, the claims are further substantiated by a comprehensive and thorough set of experiments. The authors validate their hypotheses across a diverse range of models and data types, using targeted analyses and ablations to directly support key findings like the "adaptability gap" and the "capacity vs. agility" trade-off. The results are presented with scientific rigor, including variance bands from multiple seeds, which makes the evidence both accurate and compelling.

**Requested Changes:**

I recommend this paper for acceptance. The work presents an interesting framework and the main claims are adequately supported by the experiments. I have no critical objections, but the following suggestions would help strengthen the submission.\
1.Acknowledge the Limited Scope of Evaluation (To strengthen):\
The paper's conclusions are supported by the experiments, but this support is based on two standard, clean datasets. The work would be more robust if the authors added a brief discussion to the limitations section about the potential challenges of applying G&L to more complex, noisy, or imbalanced real-world data.\
2.Provide Context on Hyperparameter Tuning (To strengthen):\
The paper would benefit from a brief clarification on how model hyperparameters were handled. A simple statement confirming whether they were tuned for a traditional metric like final accuracy, rather than the proposed cumulative error metric, would provide important context for interpreting the results and understanding the trade-offs involved.

---

> ### Author Response · Authors · 2025-12-29
> **Author Response to Reviewer K2nJ**
>
> Thank you, we appreciate your thorough and thoughtful review, and your insights and suggestions. We have updated the paper.
>
> We agree that complex real-world data are very different from the clean, small, and standard datasets used in this paper and, as suggested, have updated the paper accordingly:
>
> - In the "Limitations and Future Work" section (Section 6.3) we now mention this explicitly, noting that reliably scaling G&L to these regimes is a natural direction for future work.
> - We added a separate point about how imbalanced classes can lead to degenerate solutions.
>
> We agree that non-standard hyper-parameters could alter the adaptability of the models. We added a statement under the "Experimental Setup" section (Section 4.5):
>
> - we state that the parameters were selected based on standard conventions for final accuracy convergence rather than explicitly tuned to minimize cumulative error
>  used, keeping the focus on the adaptability of standard architectures as they are typically instantiated.
>
> We believe that this addresses all your concerns. Thank you.

---

### Review · Reviewer_U1j3 · 2025-12-16

**Summary Of Contributions:**

This paper proposes an evaluation protocol called "Guess-and-Learn (G&L)" aimed at revisiting the adaptability of machine learning models during the "Cold-Start" phase. The authors point out that current evaluation systems overly rely on Final Accuracy or Label Efficiency (the number of labels needed to reach specific performance), thereby ignoring the "Adaptation Cost" accumulated by the model while learning from scratch. To quantify this cost, G&L defines the core metric as Cumulative Error (E_N), which is the sum of errors committed by the model during the process of sequentially labeling, predicting, and updating parameters on the entire dataset.

### **strengths**

- The paper reveals a counter-intuitive phenomenon through detailed experiments (especially the analysis in Figure 2 and Figure 5 ): in the cold-start phase, "bigger" is not necessarily "better".
- It uses the official test set as the unlabeled pool and processes the entire pool without replacement during experiments, which to a certain extent mitigates the experimental differences caused by uncertainty.

### **weaknesses**

- The paper mainly relies on MNIST and AG News as experimental platforms. These two datasets are widely considered "solved" toy problems and cannot represent the real challenges faced by modern machine learning.
- The baseline methods are outdated and ignore SOTA Active Learning strategies. This leads to the paper's conclusions being limited by "weak baseline" settings based on simple uncertainty heuristics.
- The Oracle Band (7-12 errors) proposed in the paper , while intuitive, is derived based on strong assumptions of "perfect clustering" and "one-shot generalization", lacking a more universal theoretical lower bound.
- The definition of "Cold-Start" is too narrow and does not consider Prompting and Zero-Shot. Especially for current pre-training-dominant models like BERT and CLIP, the mainstream solution for cold-start is usually Zero-Shot Prompting or In-Context Learning (ICL).

**Audience:**

Yes

**Audience Explanation:**

The submission reveals a counter-intuitive phenomenon through detailed experiments (especially the analysis in Figure 2 and Figure 5 ): in the cold-start phase, "bigger" is not necessarily "better".

**Claims And Evidence:**

No

**Claims Explanation:**

The submission relies on widely considered "solved" toy datasets like MNIST and AG News that cannot represent real modern challenges , and employs outdated baseline methods that ignore SOTA Active Learning strategies, limiting the conclusions to "weak baseline" settings based on simple uncertainty heuristics.

**Requested Changes:**

- Experiments on more challenging datasets like CIFAR-100 or GLUE. Relying solely on toy datasets makes it difficult to prove that the trade-off is truly universal.
- Include SOTA strategies like k-Center Greedy. A benchmark should be compared against the best, not just weaker baselines.
- For pretrained tracks, it would be best to add a Zero-Shot baseline. If CLIP without updates outperforms fine-tuning, that would be a crucial finding.
- Explicitly state the Oracle's limitations in the main text, noting that it might be inaccurate on datasets with high noise.

---

> ### Author Response · Authors · 2025-12-29
> **Author Response to Reviewer U1j3**
>
> Thank you for your review and your feedback, we appreciate it. We have sought to address your feedback both here and with changes in the paper.
>
> Added SOTA Baseline (K-Center Greedy):
> We agree that integrating a SOTA strategy like K-Center Greedy (Sener & Savarese, 2018) is beneficial, and we have added this across all tracks. The results are in Table 2 and discussed in Section 5.3. It worked well on distance-based models, reducing cumulative error by 17.0% for k-NN on MNIST compared to Random sampling (which was the next best performer). However, for ResNet-50, it offered only a 1.3% improvement over Confidence, and failed to improve over the existing baselines for ViT-B/16. We see this as validation of our hypothesis that in order to improve adaptability, new architectures are needed - not solely different active learning strategies - and this benchmark seeks to drive their development.
>
> Dataset scope:
> We used clean, small, standard datasets to allow for separating out model adaptability from data quality issues. We agree that expanding the evaluation to more challenging benchmarks (e.g., CIFAR-100 or GLUE) is ultimately necessary to prove the universality of these trade-offs. However, even on datasets which are considered solved like MNIST, modern Transformers (ViT-B/16) still make ~150 errors during the early phase, implying that there is still room for improvement. To address experiments on more challenging datasets, we have updated the "Limitations and Future Work" section (Section 6.3) to note that scaling G&L reliably to complex, noisy, or imbalanced regimes is a clear direction for future work.
>
> Zero-Shot and CLIP:
> We believe that while CLIP represents a current gold-standard of a fully-trained model, G&L is about adaptability. Even extremely strong Zero-Shot models will have distributions they have no notions of and would require training for, and at that point G&L would become relevant. The Pretrained-Online (PO) track is explicitly created to address this scenario: how a model with (potentially strong) priors adapts when it receives further training. While using CLIP to provide a baseline for MNIST rather than relying on our heuristic oracle floor for the dataset has an initial appeal, because CLIP will undoubtedly have MNIST within its training data, that would only be demonstrating memorization - i.e. what is possible for a fully trained model to achieve, not adaptability.
>
> Oracle Limitations:
> Regarding the Oracle's validity on noisy data, we fully agree with your intuition. The heuristic floor we provided is an estimation specifically constructed for MNIST (relying on its distinct cluster purity) and is not applicable to noisy datasets like AG News. We have updated "The Adaptability Gap" (Section 5.1) to explicitly note that noisy datasets inherently possess higher error floors. However, we argue that the adaptability gap remains significant regardless of the precise theoretical floor.
>
> Thank you.

---

### Review · Reviewer_EYSi · 2025-12-22

**Summary Of Contributions:**

This paper proposes a learning paradigm called "Guess and Learn" that integrates the pool-based active learning where one can observe the full pool of samples and selects arbitrary one sample, and the online learning where one must make immediate decisions for each selected sample and suffer from a cumulative regret. The paper considers the variants of training from scratch or training from pretrained, selecting per sample or per batch, and different query strategies including random, selecting the most confidence, and selecting the least confidence. The experiments are conducted mainly on 2 types of dataset and model combinations: MNIST and CNN, AG News and BERT. The author(s) suggest 5 key findings: (1) pretrained model may still suffer in the initial stage; (2) smaller models may learn faster in the initial stage; (3) uncertainty sampling strategy may hurt the performance from the cold start; (4) batch size and resetting or not are important in the performance; (5) no dominant combination that outperforms others in all scenarios.

**Audience:**

No

**Audience Explanation:**

1. All the ingredients of this "Guess and Learn" paradigm have already been investigated: online learning for cumulative regret and immediate decision, active learning for intentionally deciding which sample to query, tuning the batch size, training from a pretrained weight, and so on. As for the paradigm itself, the most important two ingredients are the online learning and the active learning. But that's where the problem becomes weird: for the online learning, you cannot observe the full sample pool, and you have to make a decision every time a new sample "arrives", which models some real-world scenes such as the online nature of customers in some platforms; for the pool-based active learning, you can observe the full pool and decide which one to query, and the goal of the learning focuses more on the "learner" side (finally reaching a good model and querying a label suffers a constant cost rather than related to the prediction result) rather than the "decision-maker" side (suffering a cumulative regret). The two ingredients actually contradicts with each other. I cannot imagine a real-world scene that this learning paradigm is practical. Could you please help me remind one where only this paradigm can deal with but not the online convex optimization or pool-based active learning alone?

2. The five key observations are all well-known results, and there is nothing new. We all know that pretrained models need fine-tuning and that's why fine-tuning is adopted. Also, smaller models require fewer samples and are of smaller sample complexities are known. From the area of active learning, people have already known that the uncertainty sampling suffers when the labeling budget is small and the model is trained from cold start, and that's why covering-based methods are introduced (see "Active Learning Through a Covering Lens" for just one example). As for the batch size's impact on the learning outcome, there are also extensive results on it (for example, batch size and learning rate, batch size and smoothness, etc.). As for the delayed feedback model in the online learning setting, it is also well-studied. As for the "resetting" strategy in the training, I don't quite see why you need to reset to the initial pretrained model weights after a batch for a meaningful learning paradigm; that discards all the efforts in the training. I think the reason for the "sublinear regret" in Figure 4 Right is due to the "entropy" querying strategy, and the pretrained model makes fewer mistakes on those it is confident with. Finally, I don't think the "no dominance among all" observation is new, and if the author(s) want to establish a publishable result from that perspective, they should focus on a small area with more extensive data and models.

**Claims And Evidence:**

Yes

**Claims Explanation:**

The experimental results support the above five claims, although not so completely and thoroughly. In fact, I think the author(s) are too ambitious to "define" a learning protocol for all the tasks and models while the content only contains two very limited datasets (MNIST in vision and AG News in NLP) and two very limited models (CNN and BERT). I wanted to say that the author(s) should include more experiments spanning over different datasets and models to systematically support their claims, but I think it is too unrealistic. Nevertheless, the learning protocol differs from previous literature. That is the only reason I selected "Yes" for this question.

**Requested Changes:**

N/A

---

> ### Author Response · Authors · 2025-12-29
> **Author Response to Reviewer EYSi**
>
> Title: Author Response to Reviewer EYSi
>
> We thank the reviewer for their review and appreciate the opportunity to address their feedback.
>
> Models
> The review states that the paper "only contains...two very limited models (CNN and BERT)." We believe there may be a misunderstanding regarding the scope of our evaluation: we evaluated 7 distinct architectures spanning the full complexity spectrum: k-NN, Perceptron, 3-layer CNN, ResNet-50, and ViT-B/16 for vision; and Text k-NN, Text Perceptron, and BERT-base for text. This breadth allowed us to demonstrate that simple models (Perceptron/k-NN) can outperform massive ones (ViT/ResNet) in the agile cold-start phase - highlighting a "Capacity-Agility trade-off" - which we could not have done if we had only tested CNN and BERT.
>
> Datasets
> Regarding the suggestion to expand the evaluation suite, we recognize that broader datasets are essential for a complete picture of cold-start adaptation. However, we used MNIST and AG News as clean, small, standard datasets for this initial benchmark, which allow us to separate out model adaptability from data quality issues. Broader datasets will indeed be needed to characterize cold-start adaptation more fully. G&L v1 is designed to be a clean, straightforward starting point, and we have updated the paper to be more explicit in reflecting this - in "Limitations and Future Work" (sect. 6.3), we now note that scaling G&L reliably to complex, noisy, or imbalanced regimes is a natural direction for future work.
>
> Real-World Applicability
> Regarding the observation that combining online and active learning elements seems contradictory, or that the paradigm lacks real-world utility, we respectfully highlight that there are extensively used modern annotation tools such as Prodigy or Labelbox that use this very workflow (Interactive Machine Learning combining online and active learning). There is demonstrated real-world practical utility and our protocol builds upon this approach.
>
> In a practical interactive workflow (e.g., a radiologist annotating scans), the system actively selects samples while updating its parameters to provide real-time predictive assistance. G&L models this dynamic, where Cumulative Error serves as a direct proxy for the "correction cost" - the burden of manually fixing incorrect predictions - which the user seeks to minimize.
>
> Novelty of Observations
> Regarding the criticism that the key observations are well known, we agree that several individual effects (like batch size) have appeared in prior work. However, the core contribution of this paper is an integrative evaluation designed to support a larger argument: that there is a need for architectures that are more agile than current SOTA models - making fewer errors during the early rapid-learning phase rather than solely optimizing for asymptotic capacity - and to provide a novel benchmarking tool to facilitate this.
>
> To address specific points:
>
> Uncertainty Sampling: While the limitations of uncertainty sampling in cold-start regimes may be discussed in specialized or expert literature, it remains a widely adopted default in practice. For instance, modAL, a popular Python active learning library, still uses uncertainty sampling as its hard-coded default query strategy (referencing the ActiveLearner documentation), showing that the limits of this strategy are not yet settled wisdom in the field.
>
> - Batch Size Ablations: Regarding the ablation tests for batch size, we conducted this as standard scientific due diligence. The resulting finding that large batches (k=200) can outperform intermediate ones (k=50) in the early phase is counterintuitive and contradicts the assumption that smaller batches always yield better agility. We have updated the text to emphasize that this nuance would be lost without this specific empirical benchmark.
> - Weight Reset Strategy: We agree that discarding state is counter-productive for learning. This track was set as a negative control, as opposed to the standard use of G&L. We have updated Section 5.4 (renamed to "Protocol Validation and Ablations") to clarify that this serves as a sanity check to confirm the benchmark accurately isolates the contribution of the update steps, rather than as a prescribed protocol.
>
> We do not claim that G&L replaces existing protocols, but we believe it bridges a diagnostic gap that neither online learning nor active learning addresses, specifically measuring the concrete cost of errors while learning. We believe this perspective will be of interest to users in the field as it reframes adaptation efficiency as a primary evaluation objective rather than a secondary side effect.

---

### Author Response · Authors · 2026-02-27
**Camera-Ready Version Uploaded**

The camera-ready version has now been uploaded. I thank the Action Editor and the anonymous reviewers for their constructive feedback, which helped improve the manuscript.

---

### Decision · Action_Editor_B9Yu · 2026-02-02

**Recommendation:** Accept as is

**Audience:**

Yes

**Audience Explanation:**

The proposed cumulative-error perspective is conceptually interesting and is of interest to researchers working on active or interactive learning.
Several reviewers found the proposed method clear, intuitive, and a valuable tool for studying cold-start and interactive adaptation.

However, several reviewers also questioned the practical significance and novelty of the framework, noting that many of the underlying components (online learning, active learning, and known trade-offs between model size and sample efficiency) are already well studied.

**Claims And Evidence:**

Yes

**Claims Explanation:**

The study is conducted carefully and rigorously.
The Guess-and-Learn protocol is clearly specified, the experimental setup is reproducible, and the analyses are systematic across multiple tracks, models, and acquisition strategies.
Within the scope of the considered benchmarks, the empirical results consistently support the paper’s main observations regarding cumulative error behavior.

That said, several reviewers noted that the evidence is limited in breadth:  evaluation is mainly conducted on two very small, clean, and well-studied datasets (MNIST and AG News). In my opinion it also makes little sense to draw conclusions for advanced models as resnets or ViTs looking at MNIST.

While the claims are supported for the settings studied, it would be interesting to study their generality to larger, more realistic regimes. These limitations should be clearly acknowledged.